# Sexual Victimization in LGB+ Persons in Belgium: Consequences, Help-Seeking Behavior, and Othering-Based Stress

**DOI:** 10.3390/healthcare13212744

**Published:** 2025-10-29

**Authors:** Lotte De Schrijver, Elizaveta Fomenko, Barbara Krahé, Joz Motmans, Kristien Roelens, Tom Vander Beken, Ines Keygnaert

**Affiliations:** 1International Centre for Reproductive Health (ICRH), VIORESC, Department of Public Health and Primary Care, Ghent University, 9000 Ghent, Belgium; lotte.deschrijver@ugent.be (L.D.S.); elizaveta.fomenko@ugent.be (E.F.); 2Vlaamse Vereniging voor Klinisch Psychologen, 1000 Brussels, Belgium; 3Department of Psychology, University of Potsdam, 14476 Potsdam, Germany; krahe@uni-potsdam.de; 4Centre for Sexology and Gender, Ghent University Hospital, 9000 Ghent, Belgium; joz.motmans@uzgent.be; 5Department of Human Structure and Repair, Ghent University, 9000 Ghent, Belgium; kristien.roelens@ugent.be; 6Department of Obstetrics and Gynecology, Ghent University Hospital, 9000 Ghent, Belgium; 7Institute for International Research on Criminal Policy, Department of Criminology, Criminal Law and Social Law, Ghent University, 9000 Ghent, Belgium; tom.vanderbeken@ugent.be; 8Women’s Clinic, Ghent University Hospital, 9000 Ghent, Belgium

**Keywords:** sexual victimization, LGB+ population, mental health, stigma, discrimination, othering-based stress, minority stress, help-seeking, sexual minorities, mixed-methods research, Belgium

## Abstract

**Background/Objectives**: Persons identifying as lesbian, gay, bisexual, pansexual, omnisexual, queer, questioning, fluid, asexual, or other non-heterosexual orientations (LGB+ persons) have been identified as a risk group for sexual victimization (SV), which can have long-lasting negative effects on well-being and physical, mental, sexual, and reproductive health. Othering-Based Stress (OBS)—reflecting societal processes of othering and resulting from stigma, prejudice, and discrimination—may contribute to increased vulnerability to SV and its consequences in LGB+ persons and affect help-seeking behavior following victimization. This study examines the impact of SV on LGB+ persons and their help-seeking behavior after victimization. **Methods**: Using a mixed-methods explanatory sequential design, first survey data from a nationally representative sample of the Belgian population on SV, its consequences, and subsequent help-seeking behavior were collected from 4632 individuals. Of these, 2965 participants (2601 heterosexual and 364 LGB+ individuals) experienced SV and represented the final sample for the quantitative analyses. In a second phase, in-depth interviews were conducted with 40 LGB+ victims to explore their experiences more thoroughly. **Results**: LGB+ individuals reported more negative consequences following SV than heterosexual persons, particularly regarding identity-related processes such as questioning gender expression and decreases in self-esteem. They also reported additional barriers to disclosing SV and seeking help from professional services or the police, including fears of stigma, invalidation, and concerns about professionals’ LGB+ competence. No significant differences were found between LGB+ persons who explicitly identified as belonging to a sexual minority group and those who did not, neither in the perceived consequences of SV nor in help-seeking barriers. **Conclusions**: LGB+ victims of sexual violence experienced more severe identity-related consequences and faced greater barriers to professional support than heterosexual victims. These results highlight the urgent need for trauma-informed, LGB+-inclusive services and structural policy measures to improve access to appropriate care.

## 1. Introduction

Sexual violence, defined by the World Health Organization [1] as “every sexual act directed against a person’s will, by any person regardless of their relationship to the victim, in any setting”, constitutes a serious global public health problem with long-term physical, psychological, sexual, reproductive, and socio-economic consequences [2,3,4,5,6,7,8,9]. Various forms of sexual violence exist, including sexual harassment without physical contact, sexual abuse with contact but without penetration, and attempted or completed rape [10].

Although sexual violence affects individuals across all demographic groups, persons identifying as lesbian, gay, bisexual, pansexual, omnisexual, queer, questioning, fluid, asexual, or other non-heterosexual orientations (LGB+ persons) face a disproportionate vulnerability to sexual victimization (SV) [11,12,13,14]. In addition to higher prevalence rates, SV among LGB+ persons may also have specific consequences related to identity development, self-esteem, and coping, and may intersect with minority stress processes that shape both the impact of victimization and help-seeking behavior [15,16,17,18].

Help-seeking following SV is often a complex, multilayered process influenced by factors such as acknowledgement of victimization, willingness to disclose, perceived availability of support, and prior experiences with service providers [19,20,21]. For LGB+ victims, additional barriers may arise, including fears related to disclosure, anticipated stigma, limited confidence in professionals’ sensitivity, and concerns about being confronted with heteronormative or cisnormative assumptions within care and law enforcement services [22,23,24,25,26,27,28,29,30,31,32].

The decision-making framework developed by DeLoveh and Cattaneo [19] offers a valuable model to explore the different stages involved in help-seeking. This model conceptualizes help-seeking as a dynamic, multi-phased process in which victims first recognize that victimization has occurred (labeling phase), then evaluate their needs, assess available options for support, anticipate potential outcomes of disclosure, and ultimately decide whether or not to seek help. Barriers may emerge at any point in this trajectory, and decisions are shaped not only by the characteristics of the victimization itself but also by interpersonal, contextual, and systemic influences [19]. In addition to these individual and structural barriers, some victims report emotional ambivalence toward perpetrators, particularly when the assailant is a partner or a member of the same LGB+ community. This may further complicate disclosure and help-seeking [26].

In addition to the classical minority stress framework [16], this study applies the broader concept of Othering-Based Stress (OBS) as a conceptual background to understanding how societal processes of othering may affect individuals who are perceived as different from dominant social norms [33]. Othering refers to the social process in which certain groups are positioned as fundamentally different, inferior, or deviant relative to a dominant normative center, resulting in exclusion, marginalization, and unequal power relations [33,34,35]. Whereas minority stress primarily focuses on stigma, prejudice, and internalized homophobia experienced by sexual minorities, OBS conceptualizes how multiple identity markers—including sexual orientation, gender identity, migration background, or other intersecting characteristics—may expose individuals to systemic othering. OBS integrates both distal (external discrimination, systemic exclusion) and proximal (internalized stigma, fear of disclosure, anticipated rejection) stressors, emphasizing how these stress processes operate within broader structural power mechanisms and normative frameworks of social exclusion [33]. Importantly, OBS represents a paradigmatic shift by redirecting the focus from individual-level vulnerability or group-specific characteristics toward the societal dynamics and normative power structures that actively construct and maintain othered positions. Rather than locating the problem within the minority group itself, OBS highlights how dominant societal norms and institutionalized forms of exclusion create the conditions that give rise to such stress [33].

While a substantial literature on these processes exists, most research to date is based on Anglo-American contexts. There is a need to examine these issues in other cultural settings to inform context-specific prevention and care strategies. In Belgium, the project UN-MENAMAIS, in which the current study is embedded, found that approximately 64% of the general population reported lifetime experience of SV, with substantially higher rates among LGB+ individuals (78% vs. 62% for heterosexuals) [33]. However, little is known about how LGB+ survivors in Belgium experience victimization and navigate the process of seeking help.

### The Current Study

Although a growing body of research has examined the specific vulnerability to SV in LGB+ persons, evidence on how victims in this group may differ from sexual majority victims in the experienced consequences of SV and help-seeking behavior remains limited. Moreover, as noted above, the existing evidence is overwhelmingly from North America, which hampers the understanding of how broader cultural contexts may affect both consequences and help-seeking behavior in comparison to heterosexual victims. Methodologically, there is little connection between studies using quantitative approaches to obtain representative data and studies employing a qualitative approach to provide a richer and more elaborate understanding of responses to SV in LGB+ persons.

**Hypothesis 1.** 

*Based on this analysis of the current state of knowledge, our study pursued four objectives. First, to identify the correlates and perceived consequences of sexual victimization among LGB+ victims in Belgium in a representative sample, focusing on physical, emotional, sexual, relational, and identity-related domains of impact. We hypothesized that LGB+ victims would report greater negative consequences compared with heterosexual victims (Hypothesis 1).*


**Hypothesis 2.** 

*Second, to examine help-seeking behavior following sexual victimization, with particular attention to perceived barriers and facilitators to disclosure and service utilization. We hypothesized that LGB+ victims would encounter more barriers to seeking (in)formal help compared to heterosexual victims (Hypothesis 2).*


Third, to combine quantitative and qualitative data from the same participants to create a more detailed picture of the consequences of SV as well as barriers to help-seeking.

Fourth, to apply the decision-making framework of DeLoveh and Cattaneo (2017) [19] to structure the analysis of the help-seeking process. In addition, we used the theoretical framework of othering-based stress (OBS) to contextualize these barriers and facilitators in relation to systemic and structural factors.

By combining quantitative and qualitative data within a mixed-methods design, this study offers an in-depth analysis of how stigma, othering-based stress, and sexual victimization intersect in LGB+ victims’ help-seeking processes within the Belgian context.

## 2. Materials and Methods

### 2.1. Study Design

This study employed a mixed-methods explanatory sequential design [36,37], combining a quantitative survey phase with a subsequent qualitative interview phase. This approach integrates the strengths of both methodologies, while addressing their respective limitations [38,39]. The quantitative data provided prevalence estimates and broad patterns of help-seeking, whereas the qualitative phase offered in-depth insights into participants’ experiences and decision-making processes regarding sexual victimization and help-seeking.

The development of the quantitative survey instrument and the topic guide for the qualitative study was informed by a prior critical interpretive synthesis (CIS) conducted by De Schrijver et al. [40], which reviewed the existing literature on sexual victimization in sexual minority populations. This CIS identified methodological challenges in studying these populations and guided the selection of core survey topics addressing underlying mechanisms, nature, magnitude, and consequences of SV as well as help-seeking behavior among LGB+ individuals. The quantitative phase provided statistical data on SV prevalence, needed as a basis for examining its impact and help-seeking behaviors among LGB+ persons residing in Belgium. The qualitative phase subsequently complemented and contextualized these findings through an in-depth exploration of victims’ experiences and decision-making processes.

### 2.2. Sample

#### 2.2.1. Quantitative Sample

Quantitative data were collected as part of a larger nationally representative study on sexual victimization in Belgium [41]. Participants were randomly selected from the Belgian National Register and consisted of residents aged 16 to 69 years, irrespective of sexual orientation, gender identity, or legal residency status. Using the National Register as sampling frame, a disproportionate stratified random sample was drawn to obtain equal numbers of men and women across three age groups (16–24, 25–49 and 50–69 years old).

In October–November 2019, 20,760 residents were invited by post to participate in a self-administered online survey (Qualtrics). The invitation letter contained a unique link and QR code; respondents who provided informed consent could then complete the survey anonymously. One reminder letter and the possibility of obtaining a randomly assigned 30 EUR voucher were used to enhance participation. Data were collected in two waves: Wave 1 took place between October and December 2019. To increase the sample size, a second data collection (Wave 2) was organized—after approval by the National Register once activities resumed following the first COVID-19 lockdown—and ran from September 2020 to January 2021 [41]. For the present analyses, only data related to victimization were used.

The online survey was available in Dutch, English, French, as well as in Arabic, Dari/Farsi, and Pashtu, as the larger study included respondents seeking international protection in Belgium, for whom these were the most frequently spoken languages at the time of data collection [42].

In total, 4632 participants completed the survey. Of these, 89.98% (n = 4168) identified as heterosexual. The remaining 10.02% (n = 464) identified as LGB+. Seventy-two participants (65 heterosexual and 7 LGB+) did not complete the SV-related questions, resulting in a final analytic sample of *N* of 4560 (4103 heterosexual, 457 LGB+). Of these, 2965 respondents (2601 heterosexual and 364 LGB+ individuals) reported at least one experience of SV. They are considered the total sample for the purposes of this paper.

Within the LGB+ subgroup, 52.59% (n = 244) indicated that they considered themselves to belong to a sexual minority group, with 82.38% (n = 201) reporting SV experiences. Among those LGB+ individuals who did not report belonging to a sexual minority group (n = 220, 47.41%), 74.09% (n = 163) had experienced SV [43]. More information on socio-demographic data for this sample was reported in a previous publication ([43], see also Appendix A).

#### 2.2.2. Qualitative Sample

Participants in the qualitative study were recruited via multiple strategies. First, participants in the quantitative survey who indicated they had experienced sexual victimization were asked whether they agreed to be contacted for participation in a qualitative study about their experiences.

To reach data saturation in a diverse sample, additional purposive recruitment targeted specific subgroups, including male victims, victims with a migration background, and non-cisgender victims. We specifically recruited LGB+ victims from both the main sample and the additional subgroups studied in the broader quantitative study (including the general Belgian population, older adults, transgender persons and applicants for international protection). Additional recruitment rounds were conducted using snowball sampling via contacts established during the piloting phase of the quantitative study, as well as through social media, advocacy and LGB+-support organizations, and sexual assault care centers.

Participants who agreed to be interviewed were informed about the nature, duration, and procedure of the interviews. Informed consent was obtained prior to participation. Participants were assured that they could stop the interview at any time, that participation involved no risks or costs, that insurance coverage was provided during participation, and that strict confidentiality protocols were in place to protect all personal information. Participants received no financial compensation.

Data were collected by a team of five trained interviewers with backgrounds in social and (mental) health sciences. Four interviewers identified as heterosexual cisgender women and one as a gay non-binary person.

Data collection started in January 2020 but was paused in March 2020 owing to the COVID-19 pandemic and related lockdown measures. After receiving ethical approval for an adapted procedure, data collection resumed in August 2020 and lasted until January 2021, using secure online platforms (Belnet Filesender) or telephone interviews where necessary. A blind quality check during the analysis phase confirmed that there were no systematic differences in data quality between face-to-face and remote interviews.

In total, 40 self-identifying LGB+ persons participated in the qualitative study. The sample included 11 male participants, 25 female participants, and four individuals identifying as non-binary, non-gender conforming, or gender-fluid. Regarding sexual orientation, 10 participants identified as gay or lesbian, 13 as bisexual, two as asexual, five as pansexual, one as mostly straight but also bisexual, and nine as ‘other’; most of those who self-identified as ‘other’ indicated that they were still exploring an appropriate label for their sexual identity. Participants’ ages ranged from 18 to 53 years, and five participants were born outside of Belgium.

Interviews were conducted in Dutch (n = 32), French (n = 6) and English (n = 2), depending on participant preference and interviewer language proficiency. Twelve interviews were conducted face-to-face, 27 took place online via video call, one interview was conducted by telephone, and one interview was split into two sessions: the first face-to-face and the second online.

### 2.3. Ethical Approval

The study was approved by the Commission for Medical Ethics of Ghent University Hospital/Ghent University (B670201837542). It was designed and performed in line with the principles of the Declaration of Helsinki. The study included only participants aged 16 years and older, given ethical and practical regulations related to the legal age of consent to sex in Belgium. All participants gave informed consent before initiating the online survey. Informed consent was also obtained before starting the in-depth interviews.

### 2.4. Measures

#### 2.4.1. Sexual Orientation

Sexual orientation was assessed using a single-item self-identification question offering multiple response options: heterosexual, bisexual, gay/lesbian, pan-/omnisexual, asexual, and other (open-ended). Participants who reported any non-heterosexual identity were categorized as LGB+.

To allow comparisons based on minority status, participants were also asked whether they considered themselves to belong to a minority group in Belgium (yes/no). Those who answered ‘yes’ were presented with a grid to indicate which characteristics defined their minority status. These included sexual orientation, gender identity (identifying as a [cisgender] woman [cisgender] man, trans woman, trans man or other), intersex or Differences in Sex Development (DSD) condition, religion or life philosophy, skin color, ethnicity, disability, age, or another characteristic. Multiple responses were possible.

For this paper, we focused on two subsamples within the LGB+ sample: (1) LGB+ participants who indicated that they belonged to a sexual minority group in Belgium based on their sexual orientation (i.e., sexual minority group), and (2) LGB+ participants who did not report such identification (i.e., non-sexual minority group).

#### 2.4.2. Sexual Victimization

A set of 17 questions assessed sexual victimization grouped into two categories: eight items assessing hands-off SV (e.g., sexual harassment without physical contact) and nine items assessing hands-on SV (e.g., sexual abuse with contact but without penetration, (attempted) rape). These behaviorally specific items, listed in the Appendix A, were validated across genders, sexual orientations, and cultural contexts [42,44,45,46]. Participants were first asked whether they had ever experienced each of the 17 forms of SV, using a dichotomous yes/no format. Participants who endorsed at least one of the items were categorized as victims of SV, whereas participants who answered “no” to all items were categorized as nonvictims. Each “yes” response was followed up by a question asking about the frequency of the experience in the last 12 months on a 5-point scale ranging from “never” to “daily”. For each reported incident, participants were also asked to indicate the gender of the assailant(s), their relationship to the assailant(s) and whether they had known the assailant(s).

#### 2.4.3. Consequences of SV and Help-Seeking

Self-reported consequences were measured by asking participants to indicate to what extent they had experienced a list of possible consequences after their sexual victimization. Participants rated each item on a 5-point Likert scale ranging from 1 = not at all/never to 5 = very much/all the time. The assessed consequences included emotional consequences (e.g., anger, fear, sadness, shame, guilt), physical consequences (e.g., pain, injuries, bruises), inability to perform daily activities (e.g., school, work, hobbies), questioning feelings of masculinity or femininity, decreased self-esteem, avoidance of certain places and/or people, sexual problems, and relational problems. Additionally, one item assessed whether participants felt they had become stronger as a result of the incident; this item was reverse-coded to calculate the overall mean score of negative consequences.

To assess help-seeking, participants were asked about disclosure, seeking professional help, perceived barriers and satisfaction with received care, specifically in relation to the SV incident they identified as having the greatest impact on their lives. If only one incident was reported, that incident was automatically selected.

Disclosure was assessed by asking victims “Whom did you tell about this?”, with multiple response options: Partner, parent, other family member, friend, acquaintance. Formal help-seeking was assessed with the question: “Did you seek help or advice after [the incident] from…”, again allowing multiple answers: General practitioner, mental health care worker, specialized care, helplines, support groups, sexual assault care center. For participants who indicated they had not sought formal help, reasons were explored with the question: “Which were the major reasons you did not seek help or advice from these people or services?” Response options were grouped into: (1) victim-related reasons (five items; example item: “I thought nothing could be done.”), (2) interpersonal reasons (three items; example item: “I was afraid of further violence.”), (3) accessibility reasons (two items; example item: “I didn’t know where to go.”) and (4) other reasons (open response). Victims who sought help were additionally asked: “How satisfied or dissatisfied do you feel with the support you received from these people or services?”, rated on a 5-point Likert scale ranging from 1 = very dissatisfied to 5 = very satisfied.

Police help-seeking was assessed with the question: “Did you or someone else report the incident to the police?” with four response options: (1) yes, I did; (2) no, but someone else did; (3) no, nobody did; and (4) no, and I don’t know if someone else did. For the current analyses, only the first option was considered as police help-seeking; all others were categorized as ‘no police’.

Participants who did not report to the police themselves were asked: “What were the major reasons you didn’t tell the police?” Response options were grouped into: (1) victim-related reasons (four items), (2) assailant-related reasons (1 item), (3) accessibility-related reasons (one item), (4) police-related reasons (five items), and (5) other reasons (open response). Victims who reported to the police were also asked to indicate their satisfaction with the support received from police on the same 5-point Likert scale ranging from 1 = very dissatisfied to 5 = very satisfied.

### 2.5. Data Analysis

#### 2.5.1. Quantitative Analysis

Statistical analysis was performed using SPSS Statistics 26 and R 4.1.1. Descriptive statistics present the socio-demographic characteristics, sexual victimization, consequences following victimization, and help-seeking behavior of the sample.

Differences in continuous variables between participants who self-identified as heterosexual and those who self-identified as LGB+ were examined using *t*-tests for independent samples. The Levene’s test was used to check for homogeneity of variance, which led to the use of the Welch *t*-test statistic as equal variances could not be assumed. Nominal and categorical variables were compared using a Chi-square test or a Fisher’s exact test if the assumptions of the Chi-square test were not met. *p*-values were corrected for multiple testing where appropriate. Effect sizes were estimated by comparing Cramer’s V coefficient (V) for nominal variables and Hedges’ g (g) for continuous variables.

SV variables were grouped into hands-off (eight items) and hands-on SV (nine items), the latter being further grouped into sexual abuse (four items) and attempted or completed rape (five items). A detailed overview of the SV outcome measures can be found in the Appendix A. Dichotomous variables were created for all SV items to assess lifetime sexual victimization.

For the barriers to contacting professionals and the police, dichotomous variables (mentioned/not mentioned) were created for each item to assess help-seeking behavior following sexual victimization.

#### 2.5.2. Qualitative Analysis

All interviews were audio-recorded and subsequently transcribed verbatim. To ensure transcription accuracy, each transcript was reviewed and verified by a second researcher not involved in the initial transcription process, checking both the audio recording and the transcript text. This validation step was completed prior to the start of the coding process.

Based on this initial familiarization with the data through reading and listening, a first set of thematic elements was identified and used to inform the initial development of the codebook. The coding process thus combined deductive elements derived from the topic guide and the existing literature [40] with inductive insights emerging from the initial transcripts.

Data analysis followed Braun and Clarke’s thematic analysis approach [47], using NVivo 12 software for data management and coding. Coding was performed independently by pairs of researchers, who rotated throughout the process. A team of eight researchers contributed to coding and refining the codebook collaboratively.

Coding was conducted in iterative cycles, with continuous adjustment and refinement of the codebook as new themes and subthemes emerged. Discrepancies between coders were discussed in consensus meetings to ensure consistent interpretation and coding decisions.

Triangulation was achieved through several mechanisms: integration of both quantitative and qualitative data sources, the involvement of multiple researchers in coding and analysis, and iterative discussions with the larger research team, including researchers involved in the quantitative analyses. Insights from earlier interview rounds also informed adaptations to subsequent interviews, allowing for further data saturation and verification.

This mixed-methods design yielded mutually corroborated findings across different data sources, enhancing the robustness of the inferences [48]. Complementary triangulation was applied, recognizing that multiple perspectives and data types may not necessarily yield identical understandings of the research object. Instead, by combining data from different methods and observers, a more comprehensive and nuanced picture of the research questions could be developed [49]. This approach acknowledges the positionality of researchers and the importance of integrating diverse viewpoints into the analysis.

### 2.6. Use of the Dataset in Other Publications

The national survey data used for the present analyses have also been reported in two previous publications [42,43,50,51,52,53,54]. The research questions and statistical analyses in the current manuscript are entirely distinct from those studies. In addition, further papers, not overlapping in content, are being prepared based on the same national dataset to examine general mental health in the Belgian population and in specific minority groups other than the sexual minority group.

## 3. Results

### 3.1. Prevalence of SV and Minority Status

In the quantitative sample of 4103 heterosexual participants, 63.39% (n = 2601) endorsed at least one of the victimization items. In the sample of 457 LGB+ participants, the prevalence rate was 79.65% (n = 364). In most cases, assailants were male and known to the victim; however, LGB+ individuals were approximately twice as likely to experience sexual victimization by a stranger compared to heterosexual individuals [43]. Within this LGB+ group, 55.5% (n = 201) identified themselves as belonging to a sexual minority group based on their sexual orientation, while 44.5% (n = 161) did not explicitly report such minority identification. These subgroups were used for further analyses regarding othering-based stress and subgroup comparisons.

In the qualitative phase, 40 self-identifying LGB+ victims participated in in-depth interviews. Interviewees reported victimization occurring both in public and private settings, and involving both known assailants (e.g., family members, (ex)partners, friends, colleagues) and strangers. Most incidents described involved male assailants within intimate partner relationships. In the following sections, we first present the findings from the quantitative survey, followed by the qualitative interviews. For confidentiality purposes, interview participants were assigned pseudonyms which they selected themselves and which are used when quoting them in the next sections.

### 3.2. Consequences of Sexual Victimization

Across both groups, a wide range of consequences was reported, spanning emotional, physical, relational, sexual and daily functioning domains (see Table 1). The quantitative findings revealed that LGB+ victims experienced significantly more negative consequences following sexual victimization compared with heterosexual victims. In particular, questioning one’s masculinity or femininity and experiencing a decrease in self-esteem were significantly more prevalent among LGB+ victims, both with medium effect sizes. LGB+ victims also reported more frequently that the experience had made them stronger in some way, although the effect size for this item was negligible. Overall, this pattern of findings is consistent with Hypothesis 1. No significant differences were found between LGB+ victims who identified as belonging to a sexual minority and those who did not (all *p* > 0.006); therefore, subgroup comparisons were not included in Table 1.

These consequences are well-documented among sexual violence victims in general. However, the qualitative data provided deeper insights into how, for LGB+ victims specifically, some of these consequences appeared more complex, intertwined and identity-related.

#### 3.2.1. Gender Identity and Self-Perception

Many participants described how the victimization experience triggered questions about their gender identity and self-perception of masculinity or femininity. Quantitatively, LGB+ victims more often reported questioning their gender expression compared with heterosexual victims (*p* < 0.001).

“I never really felt feminine or something, but now it’s like… I feel like I’m broken or something, that there is something wrong with me.”(Sophie, 24, cisgender woman, pansexual)

#### 3.2.2. Shame, Guilt and Self-Esteem

Participants frequently mentioned a strong impact on their self-esteem and feelings of shame or guilt, which was also confirmed by a significantly greater decrease of self-esteem among LGB+ victims (*p* < 0.001).

“I feel bad about myself because I let it happen, because I didn’t defend myself or something like that.”(Eva, 23, cisgender woman, bisexual)

#### 3.2.3. Sexual Difficulties

Sexual difficulties were commonly reported, including loss of sexual desire, discomfort during intimacy, or avoidance of sexual activity altogether.

“It’s not that I want to avoid sex, but every time it gets physical, I’m like: oops, something is going to happen now.”(Ella, 27, cisgender woman, bisexual)

#### 3.2.4. Relational Difficulties

Relational consequences were also prominent. Several participants described long-term struggles with trust in (new) intimate relationships.

“Trust is always an issue… Always. I’m always on my guard, constantly.”(Liam, 21, cisgender man, gay)

#### 3.2.5. Confusion About Sexual Development and Orientation

Some LGB+ participants described confusion about the extent to which their sexual victimization experiences may have influenced their sexual development and sexual orientation. This confusion was often directly related to the gender of the assailant, who in most reported cases was male. The same-gender nature of the abuse led some victims to question whether their sexual orientation was shaped—or even caused—by the abuse.

“[…] the struggles I did experience at some point to not confuse my sexuality with the uhm sexual experiences I had in my childhood unfortunately. Uhm, to think that the two were connected or linked or because I experienced that as a child that that automatically caused me to be uhm homosexual as an adult.”(Bram, 23, cisgender man, gay)

Angela, who reported multiple experiences of sexual victimization by male assailants, including her stepfather, father and a caregiver, described how—despite being aware of her attraction to women from an early age—she still struggled for years with the question of whether her lesbian orientation could have been shaped by these experiences and her aversion to men.

“I already knew when I was 10–11 that I liked women. That—I knew that very quickly. But it has, it has taken a really long time. I think I was already 21 or something uhm, the moment I was able to effectively cross out okay, I really don’t like men.”(Angela, 29, cisgender woman, lesbian)

#### 3.2.6. Complex Long-Term Consequences and Indirect Effects

Beyond these specific identity-related challenges, several interviewees also highlighted the complexity of disentangling short- and long-term effects of sexual victimization on their lives. For many, the coping processes themselves generated additional indirect consequences over time.

“It’s like they have broken you a bit. Your, your body, your head or I don’t know which mechanisms are different—give you, yes, conflicting feelings about exactly the same thing. What makes me angry about it is that I, I realize and feel that he has taken away my potential.”(Sophie, 35, cisgender woman, attracted to people)

Many participants emphasized the difficulty of fully grasping the overall impact of sexual victimization because it had affected them in so many different ways, and because they could not know for sure how their lives would have unfolded otherwise.

“I cannot know what I would have become without what happened. I can never say if my life would have been better or worse. It’s always there in the background somehow.” (Emma, 26, cisgender woman, bisexual)

#### 3.2.7. Coping Strategies and Maladaptive Regulation

In efforts to cope and regulate emotions, some participants resorted to maladaptive coping strategies, including attempts to control situations, self-harming behaviors and various forms of addictive behavior such as substance use, alcohol, or disordered eating.

“I started drinking and eating a lot, to push everything away. But after a while, you realize you’re just escaping, and it only makes you feel worse.”(Lotte, 30, cisgender woman, pansexual)

#### 3.2.8. Posttraumatic Growth

Although most participants emphasized the negative impact of sexual victimization, a small number also referred to personal growth following their experiences.

“I’m a bit stronger than before. I’ve been able to put things into perspective and… Yeah, to recover.”(Jules, 20, cisgender man, pansexual)

### 3.3. Help-Seeking Decisions and Behavior Organized According to the Deloveh and Cattaneo Framework (2017)

In this section, the help-seeking process of LGB+ victims is analyzed using the decision-making framework proposed by DeLoveh and Cattaneo [19], which conceptualizes help-seeking as a dynamic process unfolding across four phases: (1) awareness of need, (2) perceived options, (3) anticipated outcomes, and (4) decision to act. This structure allows integration of both quantitative data on disclosure rates and barriers, as well as qualitative data providing in-depth insights into the experiences of LGB+ victims.

Quantitative results on disclosure rates, formal help-seeking behavior and satisfaction with services are summarized in Table 2. Quantitative data on reported barriers to professional help-seeking are presented in Table 3. The qualitative data complement these findings by offering in-depth insights into the barriers and facilitators experienced by LGB+ victims throughout the help-seeking process.

#### 3.3.1. Awareness of Need

Quantitative analyses revealed that 43.16% of all sexual violence victims in the full sample (heterosexual and LGB+ victims; n = 1265) did not disclose their experiences to anyone (Table 2). This suggests that for many victims, disclosure may be delayed or absent. Non-disclosure can reflect a lack of recognizing one’s experience as sexual victimization and thus not perceiving a need for help. However, it may also reflect awareness of victimization but hesitations to disclose due to anticipated stigma, fear of negative reactions, or perceived lack of safe disclosure options (cf. infra). No significant differences were found between heterosexual and LGB+ victims in these overall informal disclosure rates (Table 2). Thus, Hypothesis 2 was not confirmed for barriers to disclosure.

In the qualitative data, several LGB+ participants reported initial difficulties recognizing their experiences as victimization. For some, sexual scripts that normalize boundary violations, particularly in same-sex contexts, contributed to minimizing the incident.

“For a long time, I didn’t even know if I could call it abuse. I thought: maybe I just misunderstood what happened.”(Sofie, 24, cisgender woman, pansexual)

“If you’re a guy and it happens with another guy, people don’t see it as real rape.”(Liam, 21, cisgender man, gay)

In some cases, internalized stigma further delayed recognition:

“You already feel that being queer makes you different. When something like this happens, you kind of think: maybe this is just part of my complicated life.”(Milan, 25, cisgender man, gay)

Several LGB+ interviewees described how, by minimizing or doubting their experience, they delayed or avoided talking about it altogether, often seeking alternative coping mechanisms.

#### 3.3.2. Perceived Options

Among victims who disclosed (both heterosexual and LGB+), 56.84% turned to informal sources: friends (54.92%), partners (31.69%), and parents (19.03%) (Table 2). Formal help-seeking was rare overall, but significantly more common among LGB+ victims (8.56%) compared with heterosexual victims (2.88%) (Table 2). This pattern is opposite to the prediction in Hypothesis 2. Police reporting was low across both groups (2.56%). No significant differences were found between LGB+ victims who identified as belonging to a sexual minority group and those who did not (Table 2).

Many LGB+ participants described doubts about whether appropriate services existed to meet their specific needs. A lack of LGB+ sensitivity, heteronormative assumptions and limited knowledge about LGB+ sexual victimization were repeatedly mentioned:

“The centers and organizations that exist, they mostly focus on women assaulted by men. I didn’t really see myself reflected in that.”(Ella, 27, cisgender woman, bisexual)

“I was afraid that the psychologist would just not understand my situation, or would minimize it because of my sexual orientation.”(Sophie, 24, cisgender woman, pansexual)

“As a trans man, I was really unsure whether they would know how to handle my case without making me feel even more uncomfortable.”(Noah, 29, transgender man, gay)

The expectation of having to educate professionals before being able to address the assault discouraged some victims:

“I didn’t want to sit there and explain what being non-binary means before I could even talk about the abuse.”(Leila, 22, non-binary, pansexual)

In several cases, participants described complex emotional bonds with the perpetrator, particularly when the assailant was a current or former intimate partner or belonged to the same LGB+ community. Some victims expressed difficulties reconciling feelings of affection or loyalty with the abuse they had suffered, which further complicated decisions to disclose or seek help.

“I still had feelings for him, even after what happened. Part of me didn’t want to ruin his life by telling anyone.”(Noah, 29, transgender man, gay)

“There aren’t that many safe spaces for people like us. I was afraid that by talking about what happened, I would make things even harder for others in the community.”(Leila, 22, non-binary, pansexual)

Quantitative data (Table 3) confirmed that among both LGB+ victims who did and did not identify as belonging to a sexual minority group, perceived lack of understanding by professionals, accessibility concerns, and lack of confidence in available services were commonly endorsed barriers to seeking formal help.

#### 3.3.3. Anticipated Outcomes

Across both heterosexual and LGB+ victims in the quantitative sample, frequently endorsed barriers included fear of not being believed, victim-blaming, self-blame, fear of negative reactions from others, and perceptions that the incident was not serious enough to warrant disclosure (Table 3). Regarding Hypothesis 2, the results indicate that LGB+ victims reported significantly more barriers to contacting professional help than heterosexual victims across several domains, including victim-related reasons (e.g., feeling embarrassed, believing nothing could be done), reasons related to others (e.g., fear of further violence), and accessibility (e.g., not knowing where to go, financial limitations). LGB+ victims also reported more barriers to contacting the police, including victim-related reasons (e.g., feeling embarrassed, feeling responsible, or uncertainty about the police response) and negative experiences with the police (e.g., fear of not being taken seriously, fear that the police would do nothing, fear that the perpetrator would be caught or punished, prior negative experiences, and feeling endangered at the police station). However, heterosexual victims reported more frequently that they did not contact professional help because they felt they did not need it, and that they did not contact the police because they perceived the incident as not severe enough.

For LGB+ victims specifically, these concerns were often amplified by anticipated homophobic or transphobic stigma. In the qualitative data, participants described multiple examples of how anticipated or past negative experiences with professionals deterred them from seeking help. Victims also described professionals as LGB+ insensitive, referring to care providers who failed to acknowledge or respect their identities and lived experiences. For example, gynecologists were reported to have conducted invasive exams without sufficient consent, and police officers were said to have asked inappropriate or queer-phobic questions during reporting procedures. Some victims even avoided disclosing their sexual orientation in healthcare settings altogether, out of fear of being dismissed or stigmatized.

“I was scared that if I would tell someone, they would say: ‘Well, if you weren’t gay, this wouldn’t have happened to you.’”(Milan, 25, cisgender man, gay)

“You’re already struggling with who you are, and then this happens. I felt like it was my own fault.”(Sophie, 24, cisgender woman, pansexual)

“I know if I had to, uh, seek emergency care or if I was stuck with something, that I’m not going to immediately mention I’m bisexual. […] Just do not say it.”(Erika, 18, bisexual gender non-conforming cisgender woman)

Fear of secondary victimization by police was particularly salient among LGB+ participants:

“I was afraid that they would not take me seriously, or that they would blame me instead of the one who did this.”(Liam, 21, cisgender man, gay)

“I didn’t want to go to the police because I didn’t want to be asked those humiliating questions again.”(Eva, 23, cisgender woman, bisexual)

#### 3.3.4. Decision to Act

Despite these multiple barriers, some LGB+ victims described positive experiences that enabled disclosure and engagement with providers of professional help. Supportive informal reactions were frequently cited as critical facilitators:

“When I finally told my best friend, she just listened and believed me. That helped me to talk to a therapist as well.”(Eva, 23, cisgender woman, bisexual)

Community-based LGB+-specific services also helped some victims feel safer seeking formal support:

“Luckily, I found out about a queer support group, and that made it feel a bit safer to start talking about it.”(Leila, 22, non-binary, pansexual)

Entering therapy allowed several victims to gradually process their experiences and better understand their coping behaviors:

“It’s really because I’ve been in therapy for a year, a year and a half now, that I’ve realized that when I show that behavior, or when I express those characteristics, that there really is a reason for that, or that there isn’t a reason but that there is a cause behind it.”(Ella, 27, cisgender woman, bisexual)

Among those who sought professional help, 59.04% reported being satisfied or very satisfied with the services received. Satisfaction with police services was substantially lower (38.67%). No significant differences in satisfaction rates were found between heterosexual and LGB+ victims (*p* > 0.05).

## 4. Discussion

The present mixed-method study examined sexual victimization (SV), its consequences, and help-seeking behavior among LGB+ victims residing in Belgium. By combining quantitative and qualitative data within an explanatory sequential design, and guided by the decision-making framework of DeLoveh and Cattaneo [19], this study provides an in-depth understanding of the interplay between SV experiences, their impact, and the help-seeking process among LGB+ victims. In addition, the findings are interpreted through the lens of Othering-Based Stress (OBS), offering a broader perspective on how stigma-related processes may shape vulnerability and coping following SV.

This study has several strengths. It is one of the first to offer mixed-methods analyses of sexual victimization and help-seeking among LGB+ persons in Belgium—a context where empirical data remain limited—based on a representative national sample which facilitated comparisons with heterosexual victims of SV. The explanatory sequential design allowed us to combine broad prevalence estimates of consequences and help-seeking strategies and barriers with in-depth qualitative insights into victims’ lived experiences. This integration yielded a richer understanding of how stigma, identity and minority stress processes intersect in shaping the consequences of sexual victimization and subsequent help-seeking behavior [13,18,29,30]. The use of the DeLoveh and Cattaneo [21] decision-making framework further facilitated a nuanced analysis of the complex, multi-phase nature of help-seeking processes following sexual victimization. An additional strength lies in the detailed qualitative accounts that revealed less frequently documented factors, such as ambivalent emotional ties to perpetrators within LGB+ communities, and concerns about further stigmatizing the community itself.

### 4.1. Consequences of Sexual Victimization Among LGB+ Victims

Confirming our first hypothesis, quantitative findings indicated that LGB+ victims reported significantly more negative consequences following SV than heterosexual victims. Differences were most pronounced regarding questioning one’s masculinity/femininity and experiencing a decrease in self-esteem, where medium effect sizes were observed. These findings align with previous research highlighting that SV may have a more complex and profound impact on LGB+ individuals, affecting identity development, self-worth, and emotional well-being [15,16,17].

The qualitative data provided additional insight into these processes. In addition to general consequences commonly observed among SV victims, such as emotional distress, relational difficulties, impaired sexual functioning, and problems in daily functioning, several themes emerged that appeared particularly salient among LGB+ victims. Some reported confusion about the interplay between their SV experiences and the development of their sexual identity, particularly when the gender of the assailant aligned with their own. Moreover, some described how coping with the direct consequences of victimization led to indirect consequences such as self-harm, substance abuse, eating difficulties, and hyper-control behaviors, as also observed in other trauma research [3,9,20].

Several participants indicated difficulties in disentangling short-term and long-term effects, often expressing that they could not fully assess how their lives—and potentially their sexual identity—might have unfolded in the absence of victimization. This illustrates the pervasive and complex nature of SV consequences on life trajectories.

Notably, some victims reported experiencing elements of post-traumatic growth, illustrating the coexistence of negative consequences and resilience processes following SV, as previously documented [55,56].

### 4.2. Help-Seeking Behavior and Barriers Among LGB+ Victims

Consistent with our second hypothesis, the quantitative findings indicated that LGB+ victims experienced more barriers to help-seeking compared to heterosexual victims. While informal disclosure rates did not substantially differ between groups, formal help-seeking remained limited for both, although it was significantly more common among LGB+ victims. At first sight, it may seem contradictory that LGB+ victims were slightly more likely to seek professional help despite perceiving more barriers. However, this may reflect the higher exposure to SV and the more severe negative consequences experienced, which may increase the likelihood of perceiving a need for professional support or meeting clinical thresholds that prompt help-seeking behavior.

Barriers reported by LGB+ victims included fear of not being believed, fear of negative reactions, concerns about professionals’ limited understanding of LGB+ issues, anticipated stigma, and previous experiences with insensitive or discriminatory responses from service providers [25,26,27].

Importantly, the qualitative data also revealed that some LGB+ victims initially struggled to recognize their experiences as sexual victimization. Especially in cases involving same-sex dynamics, sexual scripts and societal minimization of boundary violations contributed to uncertainty about whether the experiences ‘counted’ as abuse. This ambiguity often delays disclosure and help-seeking.

In line with the decision-making framework of DeLoveh and Cattaneo [19], barriers emerged across all stages of the help-seeking process. Some victims struggled to recognize their experience as SV, while others anticipated invalidating reactions, experienced internalized stigma, or worried about disclosing their sexual identity during the help-seeking process.

Further, important differences were observed in the underlying reasons for not seeking help between LGB+ and heterosexual victims. For heterosexual victims, non-disclosure was often based on an assessment of not needing formal help, while for LGB+ victims, shame, guilt, and rape myths appeared to play a more central role [21]. Both groups reported limited trust that others would believe them, take them seriously, or respond sensitively and appropriately. However, LGB+ victims also experienced more interpersonal and societal barriers related to sexual stigma and prejudice. These included concerns about the absence of supportive networks, fear of causing problems within existing social networks, and concerns related to identity concealment [15,16,17].

In addition, qualitative data revealed that for some victims, ambivalent feelings toward the perpetrator further complicated disclosure decisions. These emotions were particularly present when the assailant was a (former) intimate partner or a member of the same LGB+ community. Some victims reported struggling with feelings of loyalty or concern about contributing to additional stigma toward their own community, which discouraged them from seeking help.

Structural and organizational barriers included long waiting lists, financial difficulties, and the absence of clear directories of therapists with expertise in sexual victimization and working with LGB+ populations. The importance of these barriers is underscored by prior research demonstrating the link between limited social support and the development of othering-based stress and mental health difficulties among LGB+ persons [15,16,57,58]. As mental health problems have also been identified as risk markers for sexual victimization [33,43], these cumulative barriers may contribute to increased vulnerability to re-victimization.

Additional insights emerged from the qualitative data. Several barriers correspond to proximal and distal minority stressors as described in the minority stress model [16]. Proximal stressors included fear of further marginalization, fear of escalation, and minimization of the severity of the victimization. Distal stressors involved previous experiences of stigma, systemic discrimination, and the perception that professionals such as healthcare providers or police would not take their case seriously. Victims also described professionals as LGB+ insensitive, for example, gynecologists conducting invasive exams without sufficient consent or police officers asking queerphobic or inappropriate questions during reports of SV. These accounts illustrate how anticipated stigma and a lack of LGB+ competence among professionals can function as distinct barriers to care, reinforcing mistrust in formal services.

A further distal stressor involved emotional complexity regarding the assailant. Some victims described ongoing affectionate feelings toward the assailant, potentially reflecting the perceived limited availability of safe or affirming intimate partners in LGB+ communities [18,24]. Finally, some victims avoided disclosing victimization involving same-sex partners or assailants from within the LGB+ community itself to protect the community from additional stigma or public scrutiny.

### 4.3. Interpreting Findings Through the Lens of Othering-Based Stress

Although Othering-Based Stress (OBS) was not directly assessed in the quantitative phase, it offers a valuable framework for interpreting the observed patterns. OBS describes the cumulative stress arising from stigma, discrimination, and social exclusion targeting minority groups [33].

The greater negative consequences and increased barriers to help-seeking observed among LGB+ victims can be understood within this broader context. Exposure to OBS may exacerbate trauma responses, increase identity-related confusion, and contribute to avoidance of disclosure and formal help-seeking [16,17,33]. Anticipated invalidation, homophobic or transphobic reactions, and fear of secondary victimization by professionals reflect ongoing processes of othering.

Specifically, several victims described prior negative experiences with law enforcement or healthcare professionals, including fears of queerphobic questioning, minimization of their victimization, or feeling unsafe during police reporting procedures. These fears reflect both distal and proximal stressors central to OBS and further discourage engagement with formal support structures.

Importantly, no significant differences were found between LGB+ persons who identified as belonging to a sexual minority group and those who did not, suggesting that exposure to othering processes may exert effects independently of the degree to which individuals explicitly adopt a minority identity label [17,59].

### 4.4. Limitations

Some limitations should be noted with the collection, analysis, and presentation of the data. First, despite the representative sampling procedure based on the national register, differences in participation rates meant that the collected quantitative data were not fully balanced with regard to an equal distribution over the different regions in Belgium (see [43]). In addition, the survey did not allow us to disaggregate the data based on the regions where participants lived and were most likely to seek help when needed. As there are cultural differences between Flanders, Brussels, and the Walloon region, and prevention and healthcare are regionally organized in Belgium, having a well-balanced sample stemming from, and representative of, the three regions would strengthen future studies.

In addition, the sample was not equally distributed over the different educational levels, with an overrepresentation of more advanced educational levels. This may have affected the findings as more advanced educational levels have been associated with presenting more health literacy and help-seeking behavior [60].

Moreover, sexual orientation and minority status were assessed via a single self-identification item, which may not fully capture the fluid and intersecting nature of sexual and gender identities [17,59]. The qualitative sample, while diverse in terms of gender identity and sexual orientation, may reflect selection bias, as individuals willing to share their experiences might differ from those who chose not to participate.

Furthermore, there were also limitations when processing the qualitative data. To reduce interpretation bias, the goal had been to code each interview twice by two different researchers. Due to timing issues, a few interviews have only been coded once, which might cause an interpretation bias in the data.

Finally, while the current study focused on sexual orientation minorities, the intersectional experiences of LGB+ persons with additional marginalized identities (e.g., migration background, disability, socioeconomic status) warrant further exploration in future research.

## 5. Further Directions and Conclusions

This mixed-method study provides valuable insights into the consequences of sexual victimization and the help-seeking processes among LGB+ persons in Belgium, including heterosexual persons as a comparison group. The key findings, practical implications, and recommendations are summarized in Table 4. The findings suggest that sexual victimization among LGB+ individuals is associated with more severe and complex consequences compared to heterosexual victims, with identity-related processes playing an important amplifying role. In particular, issues such as self-esteem deterioration, identity confusion, maladaptive coping, and concerns about community stigma were highlighted.

At the same time, the study underscores the complex and often fragmented nature of help-seeking processes among LGB+ victims. Although some victims did reach out for support, multiple barriers were named as operating throughout the help-seeking trajectory, ranging from difficulties in recognizing victimization, internalized stigma, anticipated invalidation, to structural and organizational obstacles. Ambivalent emotional ties to perpetrators and concerns about contributing to further marginalization of the LGB+ community emerged as additional factors complicating disclosure and formal help-seeking.

By applying the decision-making framework of Deloveh and Cattaneo [19] and interpreting findings through the lens of Othering-Based Stress [33], this study highlights the cumulative effects of stigma-related stressors that operate across multiple levels. The results emphasize the need for LGB+-inclusive prevention, culturally competent support services, and structural policy interventions addressing both individual and systemic barriers to care.

Ultimately, reducing the burden of sexual victimization and improving access to appropriate care for all victims of sexual violence will require integrated efforts across research, clinical practice, public health, education, and policy domains. Continued attention to the specific realities and needs of sexual minority populations remains crucial to ensure safe and affirming pathways to disclosure, support, and recovery.

## Figures and Tables

**Table 1 healthcare-13-02744-t001:** Consequences of sexual victimization by heterosexual and LGB+ victims.

Constructs (All Ranges 1 to 5)	Total Sample (*N* = 2965)		t; df; *p*-Value; g
	Heterosexual (*n* = 2601; Valid %: 87.72) Mean (SD)	LGB+ (*n* = 364; Valid %: 12.28) Mean (SD)	
Emotional consequences ^a^	1.95 (1.14)	2.47 (1.31)	−7.09; 435.15; <0.001; 0.445
Physical consequences ^b^	1.10 (0.41)	1.28 (0.73)	−4.62; 387.76; <0.001; 0.397
Daily life ^c^	1.14 (0.51)	1.30 (0.73)	−4.10; 407.07; <0.001; 0.301
Questioned masculinity/femininity	1.17 (0.55)	1.54 (0.94)	−7.17; 390.50; <0.001; 0.600
Decreased self-esteem	1.51 (0.93)	2.11 (1.28)	−8.44; 411.27; <0.001; 0.602
Became stronger (reversed)	1.80 (1.13)	2.00 (1.23)	−2.98; 444.08; 0.003; 0.180
Avoidant behavior ^d^	1.73 (1.14)	2.16 (1.33)	−5.82; 432.60; <0.001; 0.370
Sexual problems	1.18 (0.62)	1.48 (0.98)	−5.58; 397.42; <0.001; 0.440
Relational problems	1.25 (0.72)	1.63 (1.14)	−6.11; 396.31; <0.001; 0.486

Note. Because the comparisons in this table involved nine independent tests, we adopted a Bonferroni-corrected significance level of 0.05/9 = 0.006 for these analyses. Equal variances could not be assumed, resulting in Welch *t*-test statistics. Abbreviation: LGB+ = Lesbian, Gay, Bisexual, pan-/omnisexual, asexual, other; g = Hedges’ g.; ^a^ Emotional consequences = anger, fear, sadness, shame, guilt, etc.; ^b^ Physical consequences = pain, injuries, bruises, etc.; ^c^ Daily life = not being able to perform daily activities such as school, work, hobbies etc.; ^d^ Avoidant behavior = avoiding specific places or persons.

**Table 2 healthcare-13-02744-t002:** Disclosure and help-seeking after sexual victimization.

Item	Within Total SV Sample (*N* = 2931)			Within LGB+ Group (*n* = 362)		
	Heterosexual (*n* = 2569; 87.65%) *n* (Valid %)	LGB+ (*n* = 362; 12.35%) *n* (Valid %)	χ^2^; df; *p*-Value; V	Sexual Minority (*n* = 201; 55.52%) *n* (Valid %)	Non-Sexual Minority (*n* = 161; 44.48%) *n* (Valid %)	χ^2^; df; *p*-Value; V
**Disclosure** ^1,2^	**1458 (56.75)**	**208 (57.46)**	**0.06; 1;** **0.800; 0.004**	**124 (61.69)**	**84 (52.17)**	**3.31; 1;** **0.069; 0.096**
Partner	468 (32.10)	60 (28.85)		37 (29.84)	23 (27.38)	
Parent	264 (18.11)	53 (25.48)		29 (23.39)	24 (28.57)	
Other family member	188 (12.89)	28 (13.46)		12 (9.68)	16 (33.33)	
Friend	797 (54.66)	118 (56.73)		75 (60.48)	43 (51.91)	
Acquaintance	233 (15.98)	31 (14.90)		18 (14.52)	13 (15.48)	
**Professional help** ^1^	**74 (2.88)**	**31 (8.56)**	**29.67; 1;** **<0.001; 0.101**	**14 (6.97)**	**17 (10.56)**	**1.47; 1;** **0.225; 0.064**
General practitioner	16 (21.62)	12 (38.71)		7 (50.00)	5 (29.41)	
Mental healthcare worker	54 (72.97)	24 (77.42)		11 (78.57)	13 (76.47)	
Specialized care	6 (8.11)	3 (9.68)		1 (7.14)	2 (11.77)	
Helplines	7 (9.46)	3 (9.68)		2 (14.29)	1 (5.88)	
Support groups	5 (6.76)	1 (3.23)		0	1 (5.88)	
Sexual Assault Care Center	2 (2.70)	2 (6.45)		2 (14.29)	0	
**Police**	**60 (2.34)**	**15 (4.14)**		**8 (3.98)**	**7 (4.35)**	

Notes: Values presented in **bold** represent the overall proportions within each main subsection (Disclosure, Professional help, and Police). The χ^2^ values refer to the comparisons highlighted in bold. Abbreviation: LGB+ = Lesbian, Gay, Bisexual, pan-/omnisexual, asexual, other; V = Cramer’s V.; ^1^ Participants could indicate multiple options; therefore, the total is >100%.; ^2^ Disclosure = disclosure prior to the interview.

**Table 3 healthcare-13-02744-t003:** Barriers to help-seeking after sexual victimization.

Item	Within Total Sample (*N* = 2734)			Within LGB+ Group (*n* = 269)		
	Heterosexual *n* = 2415; (88.33%) *n* (Valid %)	LGB+ *n* = 319; (11.67%) *n* (Valid %)	χ^2^; df; *p*-Value	Sexual Minority *n* = 181; (56.74%) *n* (Valid %)	Non-Sexual Minority *n* = 138; (43.26%) *n* (Valid %)	χ^2^; df; *p*-Value
**BARRIERS TO CONTACTING PROFESSIONAL HELP (n = 2734)**						
**Reasons linked to the victim**	**2181 (90.31)**	**276 (86.52)**	**4.45; 1; 0.035**	**160 (88.40)**	**116 (84.06)**	**1.26; 1; 0.264**
I didn’t need help.	1836 (76.03)	203 (63.63)	22.81; 1; <0.001	119 (65.75)	84 (60.87)	0.80; 1; 0.370
I thought nothing could be done.	131 (5.42)	34 (10.66)	13.61; 1; <0.001	17 (9.39)	17 (12.32)	0.70; 1; 0.401
I felt embarrassed about what happened.	255 (10.56)	53 (16.61)	10.34; 1; 0.001	30 (16.58)	23 (16.67)	0.00; 1; 0.982
I would not be believed or taken seriously.	134 (5.55)	25 (7.84)	2.69; 1; 0.101	16 (8.84)	9 (6.52)	0.58; 1; 0.445
I didn’t trust anyone.	80 (3.31)	20 (6.27)	6.99; 1; 0.008	13 (7.18)	7 (5.07)	0.59; 1; 0.441
**Reasons linked to others**	**139 (5.76)**	**30 (9.40)**	**6.47; 1; 0.011**	**18 (9.94)**	**12 (8.70)**	**0.14; 1; 0.705**
I was afraid of further violence.	19 (0.79)	9 (2.82)	11.51; 1; <0.001	8 (4.42)	1 (0.72)	3.90; 1; 0.048
I didn’t want the person who did this to me to get in trouble.	97 (4.02)	13 (4.08)	0.00; 1; 0.960	6 (3.32)	7 (5.07)	0.62; 1; 0.431
I didn’t want to bring a bad name to the family/group I belong to.	49 (2.09)	10 (3.14)	1.63; 1; 0.201	4 (2.21)	6 (4.35)	1.18; 1; 0.278
**Reasons linked to accessibility**	**85 (3.52)**	**28 (8.78)**	**19.66; 1; <0.001**	**11 (6.08)**	**17 (12.32)**	**3.81; 1; 0.051**
I didn’t know where to go.	79 (3.27)	23 (7.21)	12.17; 1; <0.001	9 (4.97)	14 (15.15)	3.13; 1; 0.077
I wasn’t able to go due to financial or transportation limitations.	11 (0.46)	6 (1.88)	9.26; 1; 0.002	2 (1.11)	4 (2.90)	1.36; 1;0.243
**Other reasons**	**211 (8.74)**	**45 (14.11)**	**9.57; 1; 0.002**	**23 (12.71)**	**22 (15.94)**	**0.68; 1; 0.411**
**BARRIERS TO CONTACTING POLICE** **(N = 2839)**	**2493 (87.81)**	**346 (12.19)**		**193 (55.78)**	**153 (44.22)**	
**Reasons linked to the victim**	**2061 (82.67)**	**271 (78.32)**	**3.91; 1; 0.048**	**157 (81.35)**	**114 (74.51)**	**2.35; 1; 0.125**
It was not severe enough.	1852 (74.29)	222 (64.16)	15.83; 1; <0.001	128 (66.32)	94 (61.44)	0.88; 1; 0.347
I felt embarrassed about what happened.	222 (8.90)	64 (18.50)	30.86; 1; <0.001	39 (20.21)	25 (16.34)	0.85; 1; 0.357
I felt partly responsible for what had happened.	157 (6.30)	48 (13.87)	26.02; 1; <0.001	31 (16.06)	17 (11.11)	1.75; 1; 0.186
I did not know what would happen after I told the police.	73 (2.93)	30 (8.67)	28.65; 1; <0.001	20 (10.36)	10 (6.54)	1.58; 1; 0.209
**Reasons linked to the assailant**	248 (9.95)	61 (17.63)	18.49; 1; <0.001	31 (16.06)	30 (19.61)	0.74; 1; 0.390
The one who did this to me was someone I know.						
**Reasons linked to accessibility**	35 (1.40)	6 (1.73)	0.23; 1; 0.629	3 (1.55)	3 (1.96)	0.08; 1; 0.774
It was difficult to get to the police or to contact them.						
**Reasons linked to the police**	**310 (12.43)**	**78 (22.54)**	**26.31; 1; <0.001**	**51 (26.42)**	**27 (17.65)**	**3.77; 1; 0.052**
The police would not believe me or take me seriously.	85 (3.41)	34 (9.83)	31.15; 1; <0.001	19 (9.85)	15 (9.80)	0.00; 1; 0.990
The police would not do anything.	203 (8.14)	46 (13.30)	10.08; 1; 0.001	31 (16.06)	15 (9.80)	2.90; 1; 0.089
The one who did this to me would not get caught or get punished.	117 (4.96)	35 (10.12)	17.63; 1; <0.001	23 (11.92)	12 (7.84)	1.56; 1; 0.212
I have had previous negative experiences with the police.	22 (0.88)	10 (2.89)	10.99; 1; <0.001	7 (3.63)	3 (1.96)	0.84; 1; 0.358
I felt endangered at the police.	25 (1.00)	11 (3.18)	11.49; 1; <0.001	7 (3.63)	4 (2.61)	0.28; 1; 0.594
**Other reasons**	**328 (13.16)**	**47 (13.58)**	**0.05; 1; 0.826**	**22 (11.40)**	**25 (16.34)**	**1.77; 1; 0.183**

Note. Values presented in **bold** represent the different subsections (barriers to contacting professional help and police), as well as the different groups. The different barriers to contacting professional help and the police were grouped into four and five groups, respectively. Percentages (in brackets) present the proportion of victims who did not seek help. Chi^2^ tests in the bold rows compared participants who endorsed any of the reasons in the respective category. Rows with bullet points compare the participants in the respective groups who reported the specific barrier. Participants could indicate multiple options; therefore, the total is >100% (if the different groups (bold) of barriers are summed up and if the barriers (dots) within a group are summed up). Because the comparisons in this table involved four sets of 14 independent tests, we adopted a Bonferroni-corrected significance level of 0.05/14 = 0.004 for these analyses.

**Table 4 healthcare-13-02744-t004:** Main findings, practical implications and recommendations.

Key Findings	Practical Implications	Recommendations/Solutions
Severe and identity-related consequences of sexual victimization (SV) in LGB+ victims, notably questioning masculinity/femininity and lowered self-esteem, with long-term impact.	Services must recognize identity-related distress and potential confusion about sexual orientation.	Provide trauma-informed, LGB+-inclusive counseling; integrate screening for self-esteem problems and identity concerns.
Indirect consequences such as self-harm, substance use, eating difficulties, hyper-control behaviors.	Risk of maladaptive coping requires early detection.	Routine mental-health screening and referral for substance use, eating disorders and self-harm.
Delayed recognition of SV, especially in same-sex contexts.	Awareness that SV can occur in all relationship types is essential.	Public campaigns and education to clarify that SV is not limited to heterosexual contexts.
Multiple barriers to help-seeking, such as fear of not being believed, anticipated stigma, concerns about professionals’ LGB+ competence.	Services must be visibly inclusive and trusted by LGB+ persons.	Mandatory diversity-sensitive and trauma-informed training for healthcare, mental-health workers, social workers, and police.
Structural barriers such as long waiting lists, financial hurdles, lack of clear directories of LGB+-competent professionals.	Improve accessibility and affordability of care.	Publicly funded, low-threshold sexual assault care centers and clear national directories of LGB+-competent professionals.
Community-related concerns, such as fear of adding stigma to the LGB+ community or loyalty to perpetrators within it.	Services must address community-specific fears.	Peer-support groups and community-based services to reduce fear of community backlash.
Need for broader policy action against stigma and othering increases vulnerability and blocks care.	Policy measures must reduce othering-based stress and improve care pathways.	Prevention campaigns and public health initiatives that explicitly include LGB+ perspectives; structural reforms to lower barriers to disclosure and professional care for both LGB+ and the general population.
Research gaps, such as limited intersectional and regional data.	Evidence base remains incomplete.	Longitudinal studies and ensuring regional, educational and sociodemographic representativeness; inclusion of LGB+ individuals with intersecting minority statuses.

## Data Availability

The data supporting the findings of this study contain confidential and sensitive information and therefore cannot be shared publicly. Access to the data may be granted upon reasonable request to the corresponding author, contingent upon compliance with applicable data protection regulations and approval by the relevant ethics committee.

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
