# Peer review of "Sexual Victimization in LGB+ Persons in Belgium: Consequences, Help-Seeking Behavior, and Othering-Based Stress"

_healthcare, 2025, doi:10.3390/healthcare13212744_

Round 1

Reviewer 1 Report

Comments and Suggestions for Authors

Overall, this is a good and clear paper. I have only some comments. Please find them below:

  1. It would be better to say "people" instead of "persons" in the title and the paper.
  2. Decipher LGB+ in the abstract and the paper.
  3. Indicate a cultural population/country in the abstract. The "nationally representative" is different in different countries.
  4. Please add more keywords (up to 10) to improve indexation.
  5. Lines 72, 120, 515 and in other places: remove &. Use "and".
  6. It is unclear how the recruitment process was implemented. Please describe this process in detail. Online? Paper-and-pencil? If online, how was the link distributed?  Referring to other published studies is good, but this is a separate paper, which should be clear for readers.
  7. Reimbursement? When was the quantitative study conducted? Etc.
  8. Please include a statement that other papers, which used the same data, have no overlapping analysis if it is the case. Please also indicate whether further papers will be prepared based on the same data. Please be transparent as much as possible.
  9.  The analyses done were described well. Ethical considerations should be placed near or within the Procedure section.
  10. "The sociodemographic characteristics of the quantitative sample have been described in detail elsewhere ([41], see also Supplementary Materials). The present study builds on these earlier data, focusing specifically on the subgroup of victims of sexual victimization and their help-seeking processes.". On these earlier data - what does this mean?
  11. Are constructs in Table 1 composed of several items? If yes, please calculate internal consistent reliability. If they are 1-item scales, there is no need to calculate reliability.
  12. Please use common statistical signs for standard statistical tests. E.g., g instead of G, when describing Hedges g in Table 1.
  13. I would recommend creating a table with the summarized main results, where the authors could indicate practical implications and recommendations. This will definitely improve the readability of this paper. For instance, help-seeking behavior and barriers and specific recommendations/solutions.
  14. Avoid using one-sentence paragraphs (lines 839-840).
  15. Overall, I believe that qualitative analyses can be also repented in a form of a table. It would be easier to cite participants' replies and indicate their themes. Please consider presenting these results in a table form.

Author Response

Dear Reviewer 1,

REVIEWER 1:

Comments and Suggestions for Authors

Comment: Overall, this is a good and clear paper. I have only some comments. Please find them below:

  • Reply: We sincerely thank the reviewer for the careful reading of our manuscript and the constructive feedback. Your detailed comments helped us clarify several methodological aspects and improve the overall readability of the paper. Below we provide a point-by-point response to each suggestion.

  1. Comment: It would be better to say "people" instead of "persons" in the title and the paper.
  • Reply: Both “LGB+ persons” and “LGB+ people” are grammatically correct. While “people” is more common in everyday language, “persons” is a more formal, academic term and is appropriate in the context of this manuscript. We therefore prefer to retain “LGB+ persons” throughout the paper.

  1. Comment: Decipher LGB+ in the abstract and the paper.
  • Reply: We have now spelled out the abbreviation LGB+ in the abstract (lines 23-24). The term was already defined in the main text at lines 60-61.

  1. Comment: Indicate a cultural population/country in the abstract. The "nationally representative" is different in different countries.
  • Reply: Thank you for your suggestion. We have now specified the country by adding “of the Belgian population” in the abstract (lines 32-33).

  1. Comment: Please add more keywords (up to 10) to improve indexation.
  • Reply: We have added “sexual victimization” and “LGB+ population” to the keywords, bringing the total to the maximum of 10 keywords for indexation (lines 48-50).

  1. Comment: Lines 72, 120, 515 and in other places: remove &. Use "and".
  • Reply: We have replaced all instances of “&” with “and” throughout the manuscript.

  1. Comment: It is unclear how the recruitment process was implemented. Please describe this process in detail. Online? Paper-and-pencil? If online, how was the link distributed? Referring to other published studies is good, but this is a separate paper, which should be clear for readers.
  • Reply: We have added a detailed description of the recruitment procedure and clarified the timing and context of the two data-collection waves. These additions can be found in the revised manuscript on lines 165-179.

  1. Comment: Reimbursement? When was the quantitative study conducted? Etc.
  • Reply: Thank you for pointing this out. We have added this information for the quantitative study in lines 171-179 and for the qualitative study in lines 218-223.

  1. Comment: Please include a statement that other papers, which used the same data, have no overlapping analysis if it is the case. Please also indicate whether further papers will be prepared based on the same data. Please be transparent as much as possible.
  • Reply: We have added a new subsection “Use of the dataset in other publications” at the end of the Methods section, which clarifies that prior papers using the same dataset report different analyses and outlines planned future manuscripts based on distinct research questions. See lines 376-382.

  1. Comment: The analyses done were described well. Ethical considerations should be placed near or within the Procedure section.
  • Reply: We have moved the Ethical Considerations section to appear directly after the Study design and Sample sections in the Methods. The section can now be found on lines 240-247.

  1. Comment: "The sociodemographic characteristics of the quantitative sample have been described in detail elsewhere ([41], see also Supplementary Materials). The present study builds on these earlier data, focusing specifically on the subgroup of victims of sexual victimization and their help-seeking processes.". On these earlier data - what does this mean?
  • Reply: Thank you for pointing this out. In our previous paper [41 -> now 43], we analyzed the occurrence of sexual victimization itself and reported the full sociodemographic characteristics of the quantitative sample. In the present manuscript we build on those data by examining the next stage: help-seeking behavior after victimization. To avoid repeating the same descriptive results and duplicating content across publications, we refer readers to the earlier paper [43] for detailed information on the initial victimization findings and sample characteristics. We adapted the paragraph slightly to make it more clear (lines 192-194) and restructured the results, as most of it was already mentioned in the methods section.

  1. Comment: Are constructs in Table 1 composed of several items? If yes, please calculate internal consistent reliability. If they are 1-item scales, there is no need to calculate reliability.
  • Reply: The constructs presented in Table 1 are each measured with a single item, therefore internal reliability was not calculated.

  1. Comment: Please use common statistical signs for standard statistical tests. E.g., g instead of G, when describing Hedges g in Table 1.
  • Reply: Thank you for pointing this out. We have changed all occurrences of G to g when referring to Hedges’ g, both in Table 1 and in the Methods section.

  1. Comment: I would recommend creating a table with the summarized main results, where the authors could indicate practical implications and recommendations. This will definitely improve the readability of this paper. For instance, help-seeking behavior and barriers and specific recommendations/solutions.
  • Reply: Thank you for the suggestion. We have added a table summarizing the main results with practical implications and recommendations just before the conclusion section.

  1. Comment: Avoid using one-sentence paragraphs (lines 839-840).
  • Reply: We have merged the two paragraphs to avoid a one-sentence paragraph.

  1. Comment: Overall, I believe that qualitative analyses can be also repented in a form of a table. It would be easier to cite participants' replies and indicate their themes. Please consider presenting these results in a table form.
  • Reply: We carefully considered the idea of presenting the qualitative analyses in a table. However, in qualitative research, particularly when using thematic analysis, the depth and nuance of participants’ narratives are best conveyed through carefully selected quotations embedded in the text rather than through tabular summaries. Tables can be useful for providing an overview of codes or themes, but in our case the key aim is to illustrate how and why these themes emerged and to preserve the context and richness of participants’ words. Condensing these findings into a table would risk oversimplifying complex, nuanced accounts and would not reflect best practice for presenting qualitative results in this field. For these reasons, we chose to retain the qualitative findings in narrative form rather than in a table.

Reviewer 2 Report

Comments and Suggestions for Authors

I think this is a well-researched study that quantitatively and qualitatively explores the patterns of sexual victimization among LGB+ individuals. The researchers must have worked hard to secure a sufficient number of LGB+ individuals for analysis, as a relatively large sample size was required. No major issues were identified with the research procedures or variable measurement, and the manuscript itself was also well-written. However, I believe the quality of the paper would be even higher if the manuscript could be improved upon with some additional improvements. Here's what seems like it needs improvement:

  1. The introduction should provide a more detailed overview of the current state of research on this topic and explain the extent to which this study can fill the research gap.

  1. The three objectives of the current study can be described in the main text, and the subtitle can be just as like “Research objectives” or “Research purposes”.

  1. In addition to triangulation, it seems necessary to present what procedures were used to ensure objectivity when conducting qualitative analysis.

  1. Beyond simply presenting the results of qualitative analysis descriptively, it would be better to provide a more specific and appropriate interpretation.

  1. This study would be most helpful if it provided more specific recommendations for clinical interventions and strategies to help LGB+ individuals recover from sexual harm and the resulting stress.

Author Response

Dear Reviewer 2,

REVIEWER 2:

Comments and Suggestions for Authors

Comment: I think this is a well-researched study that quantitatively and qualitatively explores the patterns of sexual victimization among LGB+ individuals. The researchers must have worked hard to secure a sufficient number of LGB+ individuals for analysis, as a relatively large sample size was required. No major issues were identified with the research procedures or variable measurement, and the manuscript itself was also well-written. However, I believe the quality of the paper would be even higher if the manuscript could be improved upon with some additional improvements. Here's what seems like it needs improvement:

  • Reply: We sincerely thank the reviewer for the thoughtful and encouraging evaluation of our work. We are grateful for the recognition of our efforts and for the constructive suggestions, which have helped us to clarify the contribution of our study and further strengthen the manuscript. Below we provide our detailed responses to each comment.

  1. Comment:The introduction should provide a more detailed overview of the current state of research on this topic and explain the extent to which this study can fill the research gap.
  • Reply: We carefully streamlined the introduction and expanded “The current study” section to highlight the contribution of the study to the existing body of knowledge.

  1. Comment:The three objectives of the current study can be described in the main text, and the subtitle can be just as like “Research objectives” or “Research purposes”.
  • Reply: See response to previous comment. The heading was changed to “The current study” and the objectives are described in the main text.

  1. Comment:In addition to triangulation, it seems necessary to present what procedures were used to ensure objectivity when conducting qualitative analysis.
  • Reply: We clarified in the manuscript: “Discrepancies between coders were discussed in consensus meetings to ensure consistent interpretation and coding decisions.” No statistical measures of inter-rater agreement were applied.

  1. Comment:Beyond simply presenting the results of qualitative analysis descriptively, it would be better to provide a more specific and appropriate interpretation.
  • Reply: Thank you for this valuable feedback. We have added a table summarizing the main qualitative and quantitative findings together with their practical implications to make the key messages clearer. In addition, we slightly restructured the Discussion to strengthen and highlight the interpretation that we had intended but which may not have come through as strongly in the original version.

  1. Comment:This study would be most helpful if it provided more specific recommendations for clinical interventions and strategies to help LGB+ individuals recover from sexual harm and the resulting stress.
  • Reply: Thank you for pointing this out. In the table summarizing the main quantitative and qualitative findings we also added a part with recommendations for clinical interventions and strategies.

Round 2

Reviewer 1 Report

Comments and Suggestions for Authors

Thank for preparing the revised version of the paper. Overall, it has been improved in a satisfactory way. Self-citations are accurate.

Please add specific conclusions in the abstract. The narration that the results underscore something described in a very general way (lines 44-47) is not a conclusion. 

Author Response

Dear reviewer 1,

Thank you for suggesting to add the most pertinent conclusions in the abstract, we have done so in the version that is uploaded now. We also corrected some errors that reoccurred in the referencing (comma's, capitals). We look forward to the positive appreciation of this revised paper.

warm wishes,

prof dr ines keygnaert